# Identification of a Sub-Clinical *Salmonella* spp. Infection in a Dairy Cow Using a Commercially Available Stool Storage Kit

**DOI:** 10.3390/ani13172807

**Published:** 2023-09-04

**Authors:** Alice Nishigaki, Renato Previdelli, James L. Alexander, Sharmili Balarajah, Lauren Roberts, Julian R. Marchesi

**Affiliations:** 1Royal Veterinary College, 4 Royal College Street, London NW1 0TU, UK; anishigaki20@rvc.ac.uk; 2Division of Digestive Diseases, Department of Metabolism, Digestion and Reproduction, Imperial College London, St. Mary’s Hospital, London W2 1NY, UK; j.alexander@imperial.ac.uk (J.L.A.); s.balarajah@imperial.ac.uk (S.B.); lauren.roberts@imperial.ac.uk (L.R.); j.marchesi@imperial.ac.uk (J.R.M.)

**Keywords:** faecal microbiome, *Salmonella* species, microbiome preservation, sub-clinical infection, solvents, cow faecal microbiome, stool storage

## Abstract

**Simple Summary:**

Sampling stools from animals is a useful way of diagnosing diseases. Faeces contain large numbers of micro-organisms, including bacteria, and, to preserve these, farmers and owners are advised to store the stool samples in a fridge at 4 °C. However, in a farm setting, access to working onsite refrigeration is not guaranteed. This means that samples can often sit at ambient temperature for several days until they are sent to the laboratory for analysis, and these may be located at significant distances from the farms. Leaving faeces at room temperature can lead to a change in bacterial composition, which can result in bacterial overgrowth and difficulty in detecting certain species for diagnosis, such as *Salmonella* (S.) species. In this study, a commercially available animal stool storage kit was tested for its ability to chemically preserve bacterial DNA, without relying on cold storage. The kit successfully preserved *Salmonella* spp. in cow stool at room temperature. The success of kits such as this could enable the accurate diagnoses of bacterial disease where cold storage is not available.

**Abstract:**

Stool sampling is a useful tool for diagnosing gastrointestinal disease in veterinary medicine. The sub-clinical disease burden of *Salmonella* spp. in cattle can become significant for farmers. However, current methods of faecal sampling in a rural setting for diagnosis are not consistently sufficient for the preservation of *Salmonella* spp. in faeces. This study evaluated the use of a commercial stool storage kit for bacterial preservation in cow faecal samples compared to unpreserved stools placed into refrigeration at different time-points. A stool sample was collected per-rectum from one apparently healthy Holstein–Freisen cow. The sample was weighed and aliquoted into two sterile Falcon tubes and into two commercial kit tubes. The aliquots were then placed into refrigeration at 4 °C at 0, 24, and 96 h after processing. One commercial kit tube was not aliquoted and remained at ambient temperature. After 2 weeks, DNA was extracted from the samples and analysed using endpoint PCR, revealing a sub-clinical infection with *Salmonella* spp. The bacterium was best preserved when the stool was stored in the commercial kit at ambient temperature and re-homogenised immediately prior to DNA extraction. The unpreserved stool did not maintain obvious levels of *Salmonella* spp. after 24 h at ambient temperature. This commercial kit should be considered for use in the diagnosis of salmonellosis in cattle.

## 1. Introduction

In the management of adult dairy cattle, there are many pathogens of concern such as *Escherichia coli*, *Campylobacter* spp., *Mycobacterium avium* subsp. *paratuberculosis* (MAP), and *Salmonella* (S.) spp. [1]. These pathogens cause gastrointestinal disease and can result in diarrhoea, fever, reduced milk yield, and, sometimes, death [1,2]. Stool sampling is a useful tool for diagnosing the carriage of these pathogens; however, the suboptimal storage of animal stool results in poor diagnosis of these enteric pathogens using methods such as faecal culture, polymerase chain reaction (PCR), and quantitative PCR (qPCR) [3].

When parasitic infections are anticipated, farmers and owners are discouraged from freezing animal faecal samples due to the deleterious effect of freezing and thawing on parasitic egg counts and accurate diagnosis [4,5,6]. Furthermore, it has been reported that freezing samples reduces the detection of MAP [7]. Instead, for microbiome analysis, veterinarians advise that samples be refrigerated at 4 °C until they can be sent to the laboratory [8,9]. However, it must be considered that, in a farm setting, there may be limited access to cold storage. Therefore, it is not unusual for samples to remain at ambient temperature for prolonged periods before they reach the laboratory for analysis, resulting in the growth of minor species in the samples and a distortion of the true composition of the microbiota [7,10,11,12,13,14]. Better methods of preserving bovine faecal microbiota are, therefore, required, particularly one that does not involve freezing the sample and is not reliant on immediate refrigeration. Limited research on the preservation of the faecal microbiota in cow stool samples exists, and, due to the clear variation between results in other species, findings from these studies cannot be extrapolated to the ruminant gut [10,11,12,13,14,15].

The literature regarding specific pathogen preservation in animal stool is also lacking. *S. enterica* is an important zoonotic pathogen, which causes salmonellosis in adult cattle [16,17]. Cattle can be asymptomatic carriers of *Salmonella* spp. resulting in difficulty identifying infected animals that are not demonstrating clinical signs but have a drop in productivity. A meta-analysis of published studies between 2000–2017 concluded that the total pooled prevalence of *Salmonella* spp. in apparently healthy dairy cattle worldwide was 9% (95%CI: 7–12%) [18]. The importance of identifying the animals carrying the pathogen becomes increasingly important, considering the evidence supporting the vertical transmission of *Salmonella* spp. [19], as well as pathogen shedding in faeces, milk, and the colostrum, which has consequences for disease spread within a farm and, also, to humans [3]. According to the United Kingdom Animal and Plant Health Agency report for 2021, *S.* Typhimurium was the most common serovar reported to the UK Health Security Agency in humans (23% of all serovars identified), and the third most common infecting cattle (12.5% of all serovars identified) [20].

Culture-based methods for the diagnosis of salmonellosis can be time-consuming and have a low sensitivity, since it has been reported that *Salmonella* spp. are quickly overgrown by other faecal organisms [1,3]. Appropriate sample handling is therefore crucial in order to reach a definitive diagnosis. More recently, PCR has been used to detect genetic material from bacteria in samples; it is thought to be quicker and more sensitive than culture-based techniques and is often used to detect pathogens that are difficult to recover from faeces [21]. In this study, *Salmonella* spp. acts as a model pathogen for the preservation and detection of bacterial DNA using endpoint PCR in animal faeces.

A commercial stool storage kit (PERFORMAbiome.GUT, DNAGenotek) is used for the collection, storage, and preservation of microbial DNA in animal stool samples and has been developed as an alternative to cold storage. Testing on dog faeces showed promising results [13], but no such research exists in cattle. Additionally, research exploring the preservation of *Salmonella* spp. in faeces has also not been undertaken using this platform. For farms that are suffering from sub-clinical infection, as well as those with more serious burdens of disease, a successful method of preserving the faecal samples as close to fresh as possible and without the use of cold storage is required for accurate diagnoses to be made.

## 2. Materials and Methods

### 2.1. Sample Collection

A faecal sample was collected from one healthy-presenting Holstein–Friesian dairy cow. The sample was obtained rectally by hand and placed into a faecal container (Excretas). This cow is later referred to in the text as the “sub-clinically infected cow” after an infection was incidentally discovered during analysis.

A sample of *Salmonella* spp.-free cow stool, as determined by endpoint PCR, obtained from a separate animal in the same herd, was collected for use in primer optimisation methods. Water-based gel lubricant was used for welfare purposes during sample collection from both cows to reduce discomfort. The Clinical Research Ethical Review Board (CRERB) at the Royal Veterinary College granted ethical approval for the collection of animal faecal samples in this study (CR-2023-012-2). Residual samples were obtained as part of normal clinical examination of the animals.

### 2.2. Sample Processing

All processing was carried out in a Microbiological Safety Cabinet. Using the volume-controlled caps, the stool was aliquoted into two stool storage kit tubes and labelled as ‘stool storage kit (SSK) refrigerated’ and ‘SSK ambient’. The stool was also aliquoted into two sterile 50 mL Falcon tubes, these tubes were labelled as ‘unpreserved frozen’ and ‘unpreserved refrigerated’ (Figure 1). The stool aliquoted per tube ranged between 891–1230 mg. Molecular-grade sterile water was added to the unpreserved refrigerated and frozen tubes in a ratio of 1 mL:100 mg stool to create a faecal slurry that could be pipetted. To replicate the commercial kits, a sterilised steel ball bearing (VWR) was also added to the Falcon tubes to aid manual stool homogenisation. All tubes were homogenised by manual shaking for 30 s each.

### 2.3. Sample Storage

An aliquot (2.5 mL) of homogenised faecal slurry was aliquoted from the unpreserved frozen and unpreserved refrigerated tubes into three sterile microcentrifuge tubes, representing three different time-points (0 h, 24 h, and 96 h). These time points indicate hours spent at ambient temperature prior to cold storage (Figure 1). As per the manufacturer’s recommendations, 250 µL of faecal-solvent slurry was aliquoted from the ‘SSK refrigerated’ tube for all time-points.

Immediately after sample processing, the 0 h unpreserved refrigerated and ‘SSK refrigerated’ aliquots were immediately refrigerated at 4 °C, and the 0 h unpreserved freezer aliquot was frozen at −20 °C. This was repeated for all 24 and 96 h aliquots at the corresponding time-points. The ‘SSK ambient’ tube was not aliquoted and remained at ambient temperature for the duration of the storage period (2 weeks), as per manufacturer recommendations (Figure 1).

### 2.4. Bacterial Culture

*S.* Typhimurium (NCTC 12023) was cultured onto nutrient agar (5 g/L peptone, 3 g/L meat extract, 12 g/L agar) and incubated aerobically at 37 °C. For liquid culture, a 10 µL loop of the culture was used to inoculate 10 mL of nutrient broth (5 g/L peptone, 3 g/L meat extract) in a 50 mL Falcon tube and was incubated in a shaking incubator at 37 °C and 200 rpm. The optical density of the broth was measured using a spectrophotometer and adjusted to OD_600_ = 1.00, 2 × 10^8^ CFU/mL using nutrient broth.

Other Enterobacteriaceae bacteria (*Klebsiella pneumoniae*, *Shigella sonnei*, *Escherichia coli*, *Enterobacter cloacae*, and *Citrobacter freundii*), used for primer optimisation, were cultured on fastidious anaerobe agar (Neogen, Lansing, MI, USA) prepared as per the manufacturer’s instructions and supplemented with 5% *v*/*v* defibrinated horse blood (Oxoid, Basingstoke, UK). These bacteria were used only for primer specificity testing purposes.

### 2.5. DNA Extraction from Stool

Two weeks after sample storage, DNA from all aliquots was extracted using the DNeasy PowerSoil Pro Kit (Qiagen, Venlo, The Netherlands) according to the manufacturer’s guidelines. An additional heating step was included, where all samples were heated at 65 °C for 10 min prior to bead beating (Bullet blender storm, 3 min at speed 8). As per the manufacturer’s recommendations, stool in the ‘SSK ambient’ tube was aliquoted immediately prior to DNA extraction. The first aliquot of 250 µL was taken without re-homogenising the kit; however, the second 250 µL aliquot was taken after re-homogensation, as per the recommendations (Figure 1). All extracted DNA was then cleaned prior to analysis using the GeneJET PCR Purification Kit (ThermoFisher Scientific, Waltham, MA, USA) as per the manufacturer’s instructions.

### 2.6. Primer Optimisation

A set of primers specific to *Salmonella* spp. were optimised prior to their use (Table 1). For specificity testing, they were tested against DNA extracted from *S.* Typhimurium and the related Enterobacteriaceae bacteria using an endpoint PCR.

The *Salmonella* spp.-free stool collected for primer optimisation was artificially spiked with *S.* Typhimurium. Then, 900 mg of stool was aliquoted into a sterile Falcon tube; molecular-grade water was added in a ratio of 1 mL:100 mg stool. Next, 1 mL of *S.* Typhimurium liquid culture (OD_600_ = 1.00, 2 × 10^8^ CFU/mL) was vortexed, and then added to the aliquoted stool. The sample was manually homogenised. DNA was then extracted from the stool and cleaned immediately after homogenisation as previously described. The primers were then tested against the extracted DNA using PCR and qPCR to confirm specificity of detection of *Salmonella* spp. within faeces.

### 2.7. Endpoint PCR

Extracted DNA from all aliquots was analysed by endpoint PCR using KAPA DNA polymerase (Roche, Basel, Switzerland) (Bio-Rad Laboratories C1000 Touch, Hercules, CA, USA) and the *Salmonella* spp.-specific primers with a final volume of 25 µL) per PCR tube (Table A1). The PCR products were mixed with a DNA stain (Novel Juice, Sigma-Aldrich, Burlington, MA, USA) and subsequently run on a 2% *w*/*v* agarose gel by electrophoresis at 100 V for 28 min with a DNA ladder mix (GeneRuler DNA Ladder Mix, ThermoFisher Scientific). A dilution series of *S.* Typhimurium DNA was also performed to establish the limit of detection (LOD). The thermal-cycling conditions are shown in Table A1 in Appendix B.

### 2.8. qPCR

Samples were analysed using qPCR (StepOnePlus™ Real-Time PCR System, Thermofisher Scientific) using the *Salmonella* spp.-specific primers and SYBR Green Master Mix (ThermoFisher Scientific) on a 96-well plate with a final volume of 25 µL per well (Table A2). The thermal-cycling conditions are shown in Table A2 in Appendix B.

## 3. Results

### 3.1. Specificity and Sensitivity Testing

Specificity testing revealed the primers were specific for the detection of *Salmonella* spp. DNA using endpoint PCR and qPCR. Primers were used against *S.* Typhimurium DNA, as well as DNA extracted from the pure bacterial cultures of the related Enterobacteriaceae. For PCR testing, positive bands were only present on the gel in the lane containing *S.* Typhimurium DNA (Figure A1). *Salmonella* spp. DNA was also successfully detected within the cow faecal sample artificially spiked with *S.* Typhimurium on PCR (Figure A2). Additionally, only *S.* Typhimurium DNA was detected from the pure bacterial culture samples and within the spiked bovine faeces using qPCR.

For sensitivity testing, a dilution series of *S.* Typhimurium DNA was carried out; the LOD was determined to be 1000 CFU/mL using endpoint PCR (Figure A3).

### 3.2. Endpoint PCR

An analysis of the DNA extracted from the non-spiked cow stool by endpoint PCR revealed positive bands indicating the presence of sub-clinical levels of *Salmonella* spp. in the faeces (Figure 2). The PCR was repeated three times.

The detection of *Salmonella* spp. in both unpreserved faecal samples decreased as time spent at ambient temperature increased. Positive PCR bands indicating *Salmonella* spp. presence were considerably less obvious by 96 h at ambient temperature (Figure 2). The ‘SSK ambient’ re-homogenised stool aliquot produced a bright positive band on endpoint PCR after being stored at ambient temperature for 2 weeks. The ‘SSK ambient’ stool aliquot that was not re-homogenised immediately prior to DNA extraction did not display the same brightness. Similarly, stool from the aliquots from the ‘SSK refrigerated’ kit kept at 4 °C did not produce obviously positive bands when using the PCR (Figure 2).

An analysis of all cow stool aliquots using qPCR did not reveal detectable levels of *Salmonella* spp. distinguishable from the negative control.

## 4. Discussion

This original study was the first to explore the preservation of bacteria in bovine faecal samples using a commercial kit. The results demonstrated that this kit was able to successfully preserve sub-clinical levels of *Salmonella* spp. DNA in faeces at ambient temperature, whereas *Salmonella* spp. DNA in unpreserved stool aliquots became undetectable over time. This commercial kit was developed as an all-in-one system for the easy collection and stabilisation of DNA in animal faecal samples, removing the need for cold storage prior to microbiome analysis in a laboratory. In this study, *Salmonella* spp. served as a model pathogen for investigating the preservation of bacterial DNA in faecal samples. The results may be applied to similar studies in the future and are particularly important when considering the preservation other bacterial pathogens that are difficult to grow, such as *Lawsonia* spp. or *Brachyspira* spp. where PCR has been established as the method of choice for detection [23,24]. The significance of the findings in this study are emphasised in the context of the limited literature within this topic. The novel findings of this study should, therefore, be regarded when considering cattle health management, particularly faecal sample storage for the diagnosis of bacterial diseases.

While research on the effect of solvents on animal stool is lacking, one study in dogs has evaluated the use of the same commercial kit in stabilising the faecal microbiota at ambient temperature. This study found that stabilised samples using this kit yielded consistent microbiota profiles, while un-stabilised samples resulted in substantial changes to the microbiota when stored at room temperature [13]. Interestingly, our study presented similar results in cow faeces, where the kit successfully preserved the faecal microbiome, enabling the detection of *Salmonella* spp. in a sub-clinically affected cow using endpoint PCR without prior enrichment steps. The cow presented as overtly healthy at the time of sample collection with no signs of gastrointestinal disease; therefore, the levels of *Salmonella* spp. detected represented a sub-clinical infection. Notably, qPCR analysis using SYBR green reagents could not distinguish levels of *Salmonella* spp. in the cow faecal samples from the negative control, indicating that endpoint PCR is more sensitive for the detection of sub-clinical levels of infection and should be the preferred method for analysis. We concluded that the poor performance of the qPCR method was due to the primers used, potentially forming self- and hetero-dimers resulting in the negative control having a Ct value of 26.86. Therefore, future work might include re-designing these primers or using a TaqMan probe to increase sensitivity and minimise artefacts.

The study was designed to reflect a likely scenario in a clinical farm setting, where faecal samples could spend periods of time at ambient temperature before being appropriately stored in the fridge. This is especially true for those farms that do not have access to working cold storage close to the handling facilities where samples are collected. The study design, mimicking this scenario, allowed us to successfully demonstrate the impact of increasing time spent at ambient temperature prior to cold storage on the preservation of sub-clinical levels of *Salmonella* spp. in unpreserved stool. The commercial stool storage kit provided a suitable alternative to cold chain storage. While there was a lack of community-profiling data available for stool stored in the commercial kits used in this study, the success of these kits in preserving low levels of *Salmonella* spp. for detection using endpoint PCR emphasises the importance of preserving bacterial DNA in faecal samples. Importantly, the commercial kit could be favoured for use in a clinical setting, as its volume-controlled collection cap ensures that stool is added to the solvent in the correct ratio, allowing immediate storage without the need for further processing in a laboratory. Standardising the sample volume is important when considering the automated processing of large numbers of samples that would be seen from large herds of livestock or from monitoring and surveillance programmes. The manufacturer’s recommendations instructed that the kits were kept between 15 °C and 25 °C and re-homogenised immediately prior to DNA extraction. When these conditions were not followed (the ‘SSK refrigerated’, and the SSK ambient’, but not re-homogenised conditions), the positive bands indicating *Salmonella* spp. detection were not present or were not as strong compared to the stool that was stored as recommended.

### 4.1. Strengths and Limitations

This novel study was the first to investigate the use of the commercial kit stool storage kit in cow stool samples, and extensive PCR and qPCR optimisation was carried out. However, the lack of biological replicates limits the generalisability of this study within the wider dairy cattle population. Importantly, there was no secondary confirmation of sub-clinical salmonellosis in the cow.

For the purposes of this study, samples were transported back to the laboratory where the stool could be weighed and aliquoted inside a Class II biological cabinet. As the total processing time was under 6 h, it was assumed that any changes to the microbiota would have been insignificant [10,11]. However, future studies where samples are placed directly into commercial stool storage kits on the farm would be useful for replicating what would be done in a clinical setting. Storage at 4 °C was chosen to reflect the current recommendations for faecal sample storage in veterinary medicine; however, further research is required to elucidate its impact on the faecal microbiome. Despite evidence demonstrating the deleterious effect of freezing samples at −20 °C on MAP, the detection of sub-clinical levels of *Salmonella* spp. was not affected at this temperature.

### 4.2. Future Directions

The burden of sub-clinical salmonellosis in cattle should not be understated, as demonstrated by the incidental finding of a sub-clinical infection in this study. The use of the preservative commercial kits should be given consideration by veterinary practices and clients for inclusion in routine annual herd health checks. However, to make these recommendations, the next steps should include using the commercial kit to collect faeces from a larger number of animals, sampling a whole herd of apparently healthy dairy cows in a farm setting to detect levels of sub-clinical salmonellosis. Further exploration of the performance of the stool storage kit for longer-term storage of stool would also be useful. The use of a bacterial culture on unpreserved stool to serve as a comparison to PCR and qPCR techniques should be considered for future experiments, as well as pathogen-spiking studies to explore the impact of a changing microbiome on PCR and qPCR analysis. Lastly, further exploration of qPCR for the detection of low levels of *Salmonella* spp. could be carried out using a probe instead of SYBR green reagents to improve sensitivity and increase the LOD.

## 5. Conclusions

The importance of stool sample preservation in cattle was emphasised when a sub-clinical infection of *Salmonella* spp. in faecal samples was identified using a commercial stool storage kit. There was a noticeable decline in detectable levels of *Salmonella* spp. in unpreserved stool left at ambient temperature after 24-h.

In a farm setting, where there is limited access to cold storage, there may be benefit in using commercial stool storage kits to preserve enteric bacterial DNA in the faeces. However, further studies are required to investigate the performance of these kits for longer term storage of stool, as well as quantifying the burden of sub-clinical salmonellosis in dairy cattle by sampling larger numbers of animals or whole herds.

## Figures and Tables

**Figure 1 animals-13-02807-f001:**
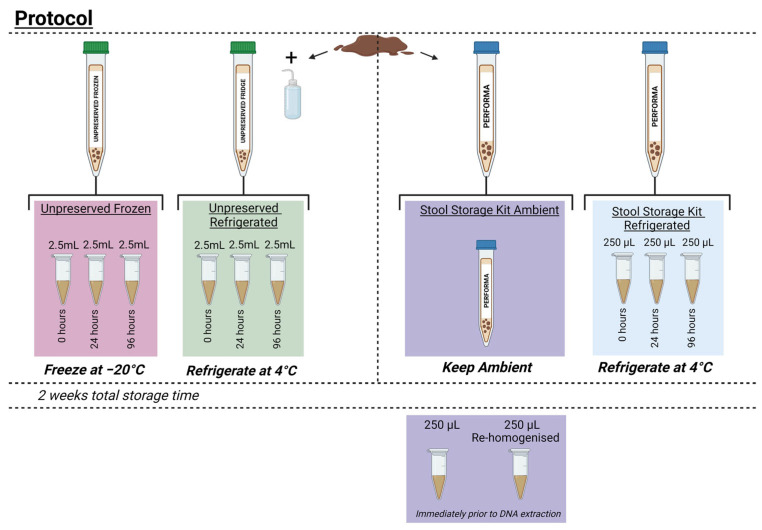
Protocol for sample processing. Stool was aliquoted into two Falcon tubes labelled ‘unpreserved frozen’ and ‘unpreserved refrigerated’, and into two commercial stool storage kits labelled ‘stool storage kit (SSK) ambient’ and ‘SSK refrigerated’. Between 891–1230 mg of stool was aliquoted per tube. Molecular-grade sterile water was added into the unpreserved stool tubes in a ratio of 1 mL water:100 mg stool to create a faecal slurry. The stool storage kit tubes contained a set volume of preservative liquid. All tubes were manually homogenised for 30 s. From the unpreserved stool in the Falcon tubes, 2.5 mL of faecal slurry was aliquoted into three microcentrifuge tubes, each to be placed into storage at different time-points (0, 24, and 96 h). Aliquots (250 µL) were taken from the SSK refrigerated tube, as per the manufacturer’s recommendations. Aliquots from the ‘unpreserved refrigerated’ and ‘SSK refrigerated’ conditions were placed at 4 °C at the specified time-points. Aliquots from the ‘unpreserved frozen’ condition were placed at −20 °C at the specified time-points. The SSK ambient tube remained at ambient temperature for two weeks and two 250 µL aliquots were taken immediately prior to DNA extraction; one aliquot was taken before the tube was re-homogenised; the other was taken after re-homogenisation as per manufacturer recommendations. Figure created in Biorender.

**Figure 2 animals-13-02807-f002:**
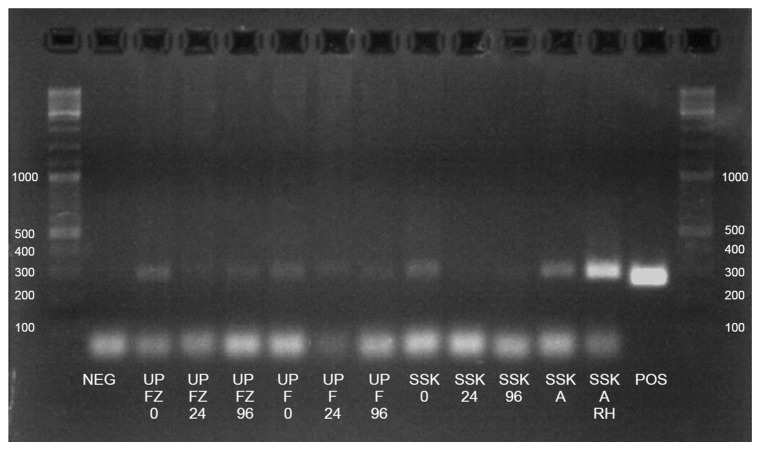
Agarose gel electrophoresis (2% *w*/*v* agarose) of PCR-amplified products from a sub-clinically affected cow faecal sample stored under different conditions using a *Salmonella*-species-specific PCR primer set. Lanes are labelled by the sample names. Abbreviations are as follows: NEG, negative control; UP FZ 0, 24, 96 = unpreserved frozen 0, 24, 96 h; UP F 0, 24, 96 = unpreserved refrigerated 0, 24, 96 h; SSK 0, 24, 96 = stool storage kit refrigerated 0, 24, 96 h; SSK A = stool storage kit ambient; SSK A RH = stool storage kit ambient re-homogenised; POS = positive control (2 × 10^8^ CFU/mL). 0, 24, and 96 indicate number of hours spent at ambient temperature before refrigeration at 4 °C or freezing at −20 °C. A DNA ladder with a range of 100–10,000 bp was placed on the end of each row (GeneRuler DNA ladder mix, ThermoFisher Scientific). Positive bands are those which are present above the negative control. Original gel images can be found in the Appendix A.

**Table 1 animals-13-02807-t001:** *Salmonella* spp. specific primers.

Name ^1^	Sequence (5′-3′)	Amplicon Size	Primer Concentration
Sal1598 Forward Primer [22]	AACGTGTTTCCGTGCGTAAT	261	0.4 pmol/µL
Sal1859 Reverse Primer [22]	TCCATCAAATTAGCGGAGGC		

^1^ The target of the primers was the *Salmonella gallinarum* invasion protein (invA) gene.

## Data Availability

No new data were created or analysed in this study. Data sharing is not applicable to this article.

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
