# Peer review of "Identification of a Sub-Clinical *Salmonella* spp. Infection in a Dairy Cow Using a Commercially Available Stool Storage Kit"

_animals, 2023, doi:10.3390/ani13172807_

Round 1
Reviewer 1 Report
Review manuscript animals-2498799
This manuscript presents a comparative study on preservation conditions for bovine fecal samples prior to molecular testing on Salmonella spp. using conventional PCR and qPCR. For this purpose, storage at room temperature for two weeks using a commercially available animal stool storage kit was compared with storage at refrigerator and freezer temperatures. The investigations show that the stool storage kit used in this study increases the sensitivity of pathogen detection considerably.
The results of this study are of interest and help for medium-term storage under field conditions for subsequent molecular analysis. These findings are also relevant for other bacterial pathogens that are difficult to cultivate and thus detection is based on molecular analysis. Furthermore, uniform storage conditions for specimen collection is an important issue for surveillance on intermittent pathogen shedding and monitoring programs.
The manuscript is written in a comprehensible and precise style.
However there are some methodological ambiguities that should be clarified and limitations of the study that should be addressed.
General Comments
The gold standard for the detection of Salmonella spp. is the bacterial culture. Bacterial culture implies the proof of viable, reproductive and infectious pathogen which is mandatory for control programs.
A comparison of pathogen detection using culture, conventional PCR and qPCR PCR would be advantageous for this study. This is especially true for the sub-clinically affected cow included in this study.
The manuscript should emphasize more clearly that Salmonella served as a model germ in this study for faecal sample storage and subsequent PCR testing. Thus, the results of this study are also relevant for other bacterial pathogens that are difficult to grow, e.g. MAP, Lawsonia or Brachyspira, and therefore PCR has been established as the method of choice for detection those pathogens.
Special comments
When Salmonella is mentioned for the first time write “Salmonella (S.)”
Replace S. typhimurium with S. Typhimurium (Typhimurium is not a species name but a serotype name)
Methodological ambiguities
The sub-clinically affected cow is not mentioned in the chapter M&M. Please add data.
The amount of Salmonella with which the faecal samples were spiked is not clearly evident. Please specify more clearly.
In the sub-chapter “Sample storage” it does not become clear under which conditions the samples „Unpreserved Frozen“, „Unpreserved Refrigerated“ and „PERFORMAbiome GUT Refrigerated“ were stored before freezing at -20°C or refrigerated at 4°C, respectively (ambient temperature? See Fig 2: “0, 24 and 96 indicate number of hours spent at ambient temperature before refrigeration at 4°C or freezing at -20°C”). If the specimens had been stored at room temperature for 0, 24 or 96 hours prior to refrigeration or freezing the samples would have been affected by two temperature conditions compared to the samples stored with the PERFORMAbiome.GUT Stool Storage kit at ambient temperature for two weeks. Is there a reason why the sample were not stored uniformly according to a simple scheme at 4°C, -20°C and at ambient temperature (PERFORMAbiome.GUT Stool Storage kit) for a period of half a week, one week, one and a half and two weeks, respectively? Why were samples store at 4°C using the using the PERFORMAbiome.GUT Stool Storage kit. The manufacturer states storage at ambient temperature for 60 days, no cold chain required.
In the sub-chapter “Bacterial culture” other Enterobacteriaceas are listed but not mentioned further in the manuscript. Were these isolates used for specificity testing of the conventional PCR and qPCR? what quantity of bacteria was used? Please clarify and provide information on the origin of these strains.
Faecal samples tested negatively for Salmonella should be spiked with a definite amount of inactivated Salmonella (not reproductive, but detectable by PCR) in order to evaluate the influence of a changing faecal mibrobiota on PCR due to different storage conditions.
In salmonella routine diagnostics faecal samples are taken and subsequently stored at room temperature, refrigerated or frozen. Therefore, it would be reasonable to store faecal samples at 4°C, -20°C and at ambient temperature up two weeks for direct comparative purposes.
qPCR has the potential to quantify the pathogen load based on a standard curve (genome equivalents). Further studies on this issue would be helpful and would allow quantification of Salmonella directly from samples.
Line 143: OD600 = 1.0; what is the corresponding CFU/ml?
Line 199-202: consider that the generation of a too large DNA product in qPCR may lead to a loss of sensitivity (s. also Chapter 4. Discussion)
Chapter 3. Results
Streamlining of this chapter is recommended (e.g. only two sub-chapters “PCR” and “sample storage”).
Leave results out of the headlines (e.g. “3.1. Specificity Testing”).
Line 211: the headline of the sub-chapter “3.1. Specificity Testing” addresses only specificity testing, but not sensitivity testing in contrast to the text. Specify and clearly separate the results for specificity and sensitivity testing in this sub-chapter. Studies on specificity and sensitivity should be performed using conventional and qPCR.
Lines 214-215, Figure 2A: the amount of Salmonella Typhimurium the cow faecal sample artificially spiked with should be given.
Line 221: Salmonella spp. were detected in the faecal sample of the non-spiked cow by conventional PCR. Was this sample also tested by qPCR?
Line 221-224: Salmonella spp. were detected in the faecal sample of the non-spiked cow by conventional PCR. Was this sample also tested by qPCR?
Figure 2: replace “from a cow” with “from a sub-clinically affected cow” in the legend.
Figure 2, Figure 1A, 2A, and 3A: please add the size of the DNA molecular weight standards of the markers for determination of the size of the PCR products.
Chapter 4. Discussion
Line 284-284: fluorescence-based exonuclease assay using a TaqMan or TaqMan MGB probe also offers the opportunity to include an internal control (duplex PCR) supplemented in the sample and or the PCR reaction mix. Please add this aspect.
A further approach to address the problem of PCR inhibitors, which are particularly prevalent in fecal samples, is to dilute the sample or the extracted DNA by a factor of 2, 5 and 10.
A product of 260 bp is unusually large for a qPCR and may lead to a loss of sensitivity. Usually qPCR using SYBR Green master is designed to produce a DNA products of about 100 bp in size.
Standardisation of sample volume and specimen vial is advantageous for automated sample processing. This aspect should be considered. This is particularly relevant for large numbers of samples (large herds of livestock, monitoring and surveillance programs).
Author Response
We would like to thank you for the evaluation of our manuscript. We have carefully read the comments and addressed the concerns raised by introducing modifications where appropriate. Please find our direct responses to your comments below.
General Comments:
The gold standard for the detection of Salmonella spp. is the bacterial culture. Bacterial culture implies the proof of viable, reproductive and infectious pathogen which is mandatory for control programs. A comparison of pathogen detection using culture, conventional PCR and qPCR PCR would be advantageous for this study. This is especially true for the sub-clinically affected cow included in this study.
Thank you for your comment. We agree that for pathogen detection, bacterial culture would be the gold standard. However, this study aimed to improve the rates of detection by using bacterial DNA as the source material, this approach would overcome those problems associated with storage of a biological viable sample, which can lead to Type II errors i.e., false negatives. By exploring the use of preservatives to fix the sample for further detection, it allowed us to use molecular techniques to assess the integrity of bacterial DNA, which ultimately addressed our aim. The authors have added a section into future directions to mitigate this concern.
The manuscript should emphasize more clearly that Salmonella served as a model germ in this study for faecal sample storage and subsequent PCR testing. Thus, the results of this study are also relevant for other bacterial pathogens that are difficult to grow, e.g. MAP, Lawsonia or Brachyspira, and therefore PCR has been established as the method of choice for detection those pathogens.
Thank you for the helpful feedback, we agree, and it has now been emphasised in the introduction and discussion.
Special comments
When Salmonella is mentioned for the first time write “Salmonella (S.)” Replace S. typhimurium with S. Typhimurium (Typhimurium is not a species name but a serotype name)
Thank you for the comment, this has been edited accordingly.
Methodological ambiguities
The sub-clinically affected cow is not mentioned in the chapter M&M. Please add data.
Thank you for the comment. The cow which was first mentioned in materials and methods was sub clinically infected. We acknowledge this accidental omission, and a sentence has been added to clarify this fact.
The amount of Salmonella with which the faecal samples were spiked is not clearly evident. Please specify more clearly.
Thank you for the comment. To address this, we included the CFU/ml in the methods when spiking is discussed.
In the sub-chapter “Sample storage” it does not become clear under which conditions the samples „Unpreserved Frozen“, „Unpreserved Refrigerated“ and „PERFORMAbiome GUT Refrigerated“ were stored before freezing at -20°C or refrigerated at 4°C, respectively (ambient temperature? See Fig 2: “0, 24 and 96 indicate number of hours spent at ambient temperature before refrigeration at 4°C or freezing at -20°C”).
Thank you for the comment. We have included that samples remained at ambient temperature until they were placed into cold storage (4°C or -20°C) after 0, 24 or 96-hours. The 0-hour samples therefore did not spend any time at ambient temperature after processing.
If the specimens had been stored at room temperature for 0, 24 or 96 hours prior to refrigeration or freezing the samples would have been affected by two temperature conditions compared to the samples stored with the PERFORMAbiome.GUT Stool Storage kit at ambient temperature for two weeks.
Thank you for highlighting this important point. This was considered by the authors prior to submission, and it was for this reason that an additional PERFORMAbiome.GUT refrigerated condition was included, despite manufacturer recommendations indicating that cold chain storage is not required. The PERFORMAbiome.GUT refrigerated condition was included so that stool stored in this kit would also be affected by two temperature conditions, like the unpreserved stool. We confirmed that by exposing the stool in the kit to both ambient and cold temperatures, detection of Salmonella spp. by end-point PCR was not possible after 24-hours at ambient temperature, unlike in the kit that remained at ambient temperature for 2 weeks and Salmonella spp. DNA could still be recovered. While it would have been useful to include an additional unpreserved condition where the stool remained at ambient temperature for 2-weeks like the PERFORMAbiome.GUT ambient condition, this was not done as in a clinical setting we determined that it would be unlikely for a sample to remain at ambient temperature for this long. From our veterinary experience, in a realistic farm setting, farmers would place the sample into refrigeration as advised by vets as soon as it was readily available, or even if this was not the case, the sample would be appropriately stored once it was received at the lab.
Is there a reason why the sample were not stored uniformly according to a simple scheme at 4°C, -20°C and at ambient temperature (PERFORMAbiome.GUT Stool Storage kit) for a period of half a week, one week, one and a half and two weeks, respectively? Why were samples store at 4°C using the using the PERFORMAbiome.GUT Stool Storage kit. The manufacturer states storage at ambient temperature for 60 days, no cold chain required.
Thank you for the interesting comment. The methods were designed to reflect a realistic, veterinary situation in a farm setting where samples would be collected and immediately refrigerated, or they could spend time at ambient temperature before access to appropriate cold storage on the farm was available. This is particularly true for those farms with limited access to cold storage in handling facilities where the samples are initially taken. The issue of time spent at ambient temperature is also important for the samples that are collected and immediately sent to the lab, due to issues with maintaining cold chain storage during transport. While it is unusual for samples to be recommended to be frozen, the -20°C was included to observe any negative affect of freezing the stool samples on bacterial detection, as has been reported for MAP. This study aimed, not on the effect of ambient, 4°C and -20°C storage comparatively, but on the effect of increasing time spent at ambient temperature on bacterial viability and recoverability of DNA, reflecting how long it may take a busy farmer to move the sample to the fridge after collection. Our results address this aim. As previously explained, the sample in the PERFORMAbiome.GUT refrigerated condition was stored at 4 °C to match the conditions that the unpreserved stool was kept at and observe any effects. Without this condition it would not have been possible to evaluate the success of the kit compared to the unpreserved stool, as the impact of exposing stool to both ambient and cold temperatures in the kit would have been unknown.
In the sub-chapter “Bacterial culture” other Enterobacteriaceas are listed but not mentioned further in the manuscript. Were these isolates used for specificity testing of the conventional PCR and qPCR? what quantity of bacteria was used? Please clarify and provide information on the origin of these strains.
Thank you for highlighting this. This information was clarified in the manuscript, as these were only used for primer specificity testing.
Faecal samples tested negatively for Salmonella should be spiked with a definite amount of inactivated Salmonella (not reproductive, but detectable by PCR) in order to evaluate the influence of a changing faecal mibrobiota (sic) on PCR due to different storage conditions.
Thank you for the comment. For the purposes of this study, artificial spiking was only carried out to obtain DNA from a faecal sample containing Salmonella, so that it could be used to confirm primer specificity for Salmonella, which addressed the aim of this study. However, we agree with the point raised, and we acknowledge this limitation. Additional experiments exploring the detection of spiked Salmonella within cow faecal samples were undertaken in relation to the current study, however further exploration is needed to validate these results. These results fall slightly out of the scope of this short communication which was written with a veterinary focus on preserving un-spiked levels of Salmonella that represented a sub-clinical infection, emphasising the potential use of commercial kits on farms to preserve bacterial pathogens in faeces. Certainly, there is great benefit in more spiking studies to explore the effect of a changing faecal microbiota on PCR to be applied to clinically infected animals. However, the main focus of our study was the benefit in using a commercial kit to preserve sub-clinical levels of bacteria that would have otherwise gone undetected in a cow that was not showing any clinical signs but could have been contributing to a loss of productivity on the farm; the finding of sub clinical salmonellosis in this study was incidental, emphasising the importance of detecting sub clinical infection on farms.
In salmonella routine diagnostics faecal samples are taken and subsequently stored at room temperature, refrigerated or frozen. Therefore, it would be reasonable to store faecal samples at 4°C, -20°C and at ambient temperature up two weeks for direct comparative purposes.
Thank you for the suggestion. All aliquots in the study, as well as the PERFORMAbiome.GUT, ambient remained in storage for 2 weeks prior to DNA extraction. The only difference was the time spent at ambient temperature prior to storage at 4°C or -20°C, reflecting how access to cold storage on farms and the time taken for samples to be appropriately stored.
qPCR has the potential to quantify the pathogen load based on a standard curve (genome equivalents). Further studies on this issue would be helpful and would allow quantification of Salmonella directly from samples.
Thank you for the comment. We agree that qPCR would be helpful to quantify Salmonella. Further experiments were undertaken in relation to this study where cow stool was artificially spiked with a known quantity of Salmonella and successfully quantified by qPCR after preservation with different solvents. We chose not to present these results in the current study, as it became clear that using qPCR with SYBR green reagents, quantification of sub-clinical (un-spiked) levels of Salmonella in cow stool was not possible. As mentioned further on in the review, we agree that future exploration of pathogen preservation and quantification using fluorescence-based exonuclease assay using a TaqMan or TaqMan MGB probe would be useful for quantifying bacteria in sub-clinically affected samples, as was seen in the current study.
Line 143: OD600 = 1.0; what is the corresponding CFU/ml?
The corresponding CFU has been added to the text.
Line 199-202: consider that the generation of a too large DNA product in qPCR may lead to a loss of sensitivity (s. also Chapter 4. Discussion)
The lack of sensitivity in the qPCR is due to potential for the formation of primer dimers, which resulted in a high Ct value for the negative control. This has been addressed in the discussion and potential improvements have been suggested.
Chapter 3. Results Streamlining of this chapter is recommended (e.g. only two sub-chapters “PCR” and “sample storage”).
Completed
Leave results out of the headlines (e.g. “3.1. Specificity Testing”).
Corrected
Line 211: the headline of the sub-chapter “3.1. Specificity Testing” addresses only specificity testing, but not sensitivity testing in contrast to the text. Specify and clearly separate the results for specificity and sensitivity testing in this sub-chapter. Studies on specificity and sensitivity should be performed using conventional and qPCR.
Corrected
Lines 214-215, Figure 2A: the amount of Salmonella Typhimurium the cow faecal sample artificially spiked with should be given.
Completed
Line 221: Salmonella spp. were detected in the faecal sample of the non-spiked cow by conventional PCR. Was this sample also tested by qPCR?
Clarified
Line 221-224: Salmonella spp. were detected in the faecal sample of the non-spiked cow by conventional PCR. Was this sample also tested by qPCR?
Clarified
Figure 2: replace “from a cow” with “from a sub-clinically affected cow” in the legend.
Corrected
Figure 2, Figure 1A, 2A, and 3A: please add the size of the DNA molecular weight standards of the markers for determination of the size of the PCR products.
Corrected
Chapter 4. Discussion Line 284-284: fluorescence-based exonuclease assay using a TaqMan or TaqMan MGB probe also offers the opportunity to include an internal control (duplex PCR) supplemented in the sample and or the PCR reaction mix. Please add this aspect.
We agree that this would be the ideal approach to the assay. We have addressed this limitation in the text. A further approach to address the problem of PCR inhibitors, which are particularly prevalent in fecal samples, is to dilute the sample or the extracted DNA by a factor of 2, 5 and 10. Fecal samples do often contain inhibitors, therefore fecal samples were spiked with bacteria to determine if this was the case as explained in the Method section “primer optimisation”. The products amplified well so we did not feel it was necessary to dilute the sample, and additionally this may reduce the limit of detection.
A product of 260 bp is unusually large for a qPCR and may lead to a loss of sensitivity. Usually qPCR using SYBR Green master is designed to produce a DNA products of about 100 bp in size.
In the past we have had successful results with primers this size and larger for qPCR. The size of this primer would allow the flexibility of using it for PCR and qPCR. The primers were also obtained from the FDA’s guidance documentation of detection of Salmonella, giving us confidence that it would yield a good result. The lack of sensitivity of the qPCR with these primers was determined to be due to primer dimer formation which is addressed in the discussion.
Standardisation of sample volume and specimen vial is advantageous for automated sample processing. This aspect should be considered. This is particularly relevant for large numbers of samples (large herds of livestock, monitoring and surveillance programs).
Now included in discussion.
Reviewer 2 Report
The manuscript is a brief communication describing the evaluation of a product (PERFORMAbiome.GUT) on bacterial preservation in "one cow faecal sample" in comparison to unpreserved stool placed into refrigeration at different time-points. The authors state that the kit was “able to chemically preserve bacterial DNA, without relying on cold storage. The kit successfully preserved Salmonella spp. in cow stool at room temperature. The success of kits such as this could enable the accurate diagnoses of bacterial disease where cold storage is not available."
In my opinion, the development of such product is necessary. However, this a technical development necessary for the commercialization of the product. In addition, the manuscript describes the analysis of only one sample. Therefore, the presented results did not demonstrate enough scientific evidence for the validation of the kit. Finally (and very important), the manuscript cited 41 times the name of the product ("PERFORMABiome"), including in the title. So the article seems to be more of an attempt to promote this product. I see absolutely no scientific merit in the manuscript.
Therefore I think it is not a scientific manuscript and it must be rejected. And I suggest the authors to perform more experiments to provide stronger scientific evidences to validate this product, before submitting the manuscript again.
The English language is Ok.
Author Response
We would like to thank you for the evaluation of our manuscript. We have carefully read the comments and addressed the concerns raised by introducing modifications where appropriate. Please find our direct responses to your comments below.
The manuscript is a brief communication describing the evaluation of a product (PERFORMAbiome.GUT) on bacterial preservation in "one cow faecal sample" in comparison to unpreserved stool placed into refrigeration at different time-points. The authors state that the kit was “able to chemically preserve bacterial DNA, without relying on cold storage. The kit successfully preserved Salmonella spp. in cow stool at room temperature. The success of kits such as this could enable the accurate diagnoses of bacterial disease where cold storage is not available." In my opinion, the development of such product is necessary. However, this a technical development necessary for the commercialization of the product. In addition, the manuscript describes the analysis of only one sample. Therefore, the presented results did not demonstrate enough scientific evidence for the validation of the kit. Finally (and very important), the manuscript cited 41 times the name of the product ("PERFORMABiome"), including in the title. So the article seems to be more of an attempt to promote this product. I see absolutely no scientific merit in the manuscript. Therefore I think it is not a scientific manuscript and it must be rejected. And I suggest the authors to perform more experiments to provide stronger scientific evidences to validate this product, before submitting the manuscript again.
Thank you for the constructive feedback. We agree that such tool is necessary and its use in a farm setting can significantly impact bacterial preservation in stool samples for further diagnosis. The title has been changed to ‘a commercial kit’ instead of using ‘PERFORMAbiome.GUT’. We understand the concern raised regarding the use of the kit’s name in the manuscript. Throughout this manuscript we kept all names and details as accurate as possible, and we had no intention of commercially promoting this kit. Furthermore, the authors have no affiliation with the company manufacturing the kit. Our Conflict of Interest statement will also declare that we do not seek or will not gain any financial benefit from the company that manufactures the “‘PERFORMAbiome.GUT” product. We acknowledge that this manuscript has a scientific foundation as the results obtained were consistent with other scientific evidence in the literature. Additionally, this method of bacterial preservation is innovative, and we opted to submit our results to present novel findings and their importance to a realistic, clinical setting. The literature surrounding bovine microbiome is very limited, and therefore publishing about methods to optimise sample collection may assist in the generation of additional studies and data. Nevertheless, the limited power of this study is acknowledged in the discussion; and this is reflected in the decision to try and publish as a short communication rather than a research article. We believe the findings of the success of this kit in preserving sub-clinical levels of Salmonella spp. that would have otherwise gone undetected if left at ambient temperature for over 24 hours fit the description of communication for this journal. Lastly, the number of times the name “PERFORMAbiome.GUT” kit was mentioned in the text has been reduced as suggested, from 41 to 28 mentions.
Round 2
Reviewer 1 Report
Review manuscript animals-2498799
The authors have responded appropriately to my comments and questions and given plausibly answers to the points raised.
The authors have plausibly presented the focus of the article in the form of a “Communication” and further experiments are subject to a follow-up study.
The main questions about the time of the storage of the fecal samples could be clarified.
However, this point could be explained in an even clearer way by a minor change and addition in the revised manuscript “animals-2498799 - after revision.pdf”, respectively:
First sentence of the chapter “DNA extraction from stool”: please replace “2-weeks after sample processing” with “2-weeks after sample storage”.
Replace in Figure 1 “2 weeks storage” with “2 weeks total storage time”.
Please replace “typhimurium” with “Typhimurium” (Typhimrium is not a species name but a term for a serotype). This point has not been realized in the revised version “animals-2498799 - after revision.pdf”.
Author Response
“The main questions about the time of the storage of the fecal samples could be clarified.
However, this point could be explained in an even clearer way by a minor change and addition in the revised manuscript “animals-2498799 - after revision.pdf”, respectively:
First sentence of the chapter “DNA extraction from stool”: please replace “2-weeks after sample processing” with “2-weeks after sample storage”.
This has now been changed to “Two weeks after sample storage”
Please replace “typhimurium” with “Typhimurium” (Typhimrium is not a species name but a term
for a serotype). This point has not been realized in the revised version “animals-2498799 - after
revision.pdf”.”
Thank you for your comments. The suggested changes have been made.
Reviewer 2 Report
Review comments to the editor and the authors:
The manuscript is a brief communication describing the evaluation of a product (PERFORMAbiome.GUT) on bacterial preservation in "one cow faecal sample" in comparison to unpreserved stool placed into refrigeration at different time-points.
I have already reviewed the first version of this manuscript. As I stated before, it describes a technical development necessary for the commercialization of the product. The presented results did not demonstrate minimal information for a scientific manuscript. I suggested the authors to perform more experiments to provide stronger scientific evidences to validate this product, before submitting the manuscript again. However, the authors have only performed some modifications in the text, to reduce from 41 to 28 mentions of the commercial kit. I see absolutely no scientific merit in the manuscript.
Therefore, I think it must be rejected.
Ok.
Author Response
“The manuscript is a brief communication describing the evaluation of a product (PERFORMAbiome.GUT) on bacterial preservation in "one cow faecal sample" in comparison to unpreserved stool placed into refrigeration at different time-points. I have already reviewed the first version of this manuscript. As I stated before, it describes a technical development necessary for the commercialization of the product. The presented results did not demonstrate minimal information for a scientific manuscript. I suggested the authors to perform more experiments to provide stronger scientific evidences to validate this product, before submitting the manuscript again. However, the authors have only performed some modifications in the text, to reduce from 41 to 28 mentions of the commercial kit. I see absolutely no scientific merit in the manuscript.
Cont.,
Therefore, I think it must be rejected.”
Thank you for your comments. Unfortunately, we are not able to perform more experiments as part of this study. We acknowledge the limitations of our research but believe the manuscript still comes under the scope of a short communication.